# Performance of Porous Asphalt Mixtures Containing Recycled Concrete Aggregate and Fly Ash

**DOI:** 10.3390/ma15186363

**Published:** 2022-09-13

**Authors:** Asad Elmagarhe, Qing Lu, Mohammad Alharthai, Mohammed Alamri, Ahmed Elnihum

**Affiliations:** 1Department of Civil and Environmental Engineering, University of South Florida, Tampa, FL 33620, USA; 2Department of Civil Engineering, University of Zawiya, Zawiya 16418, Libya; 3Department of Civil Engineering, Najran University, Najran 66446, Saudi Arabia; 4Department of Civil Engineering, King Saud University, Riyadh 11421, Saudi Arabia

**Keywords:** porous asphalt mixture, recycled concrete aggregate, fly ash, mix design, moisture susceptibility, Cantabro test, macrotexture, sound absorption

## Abstract

This study investigates the effects of two waste materials from construction and industry, namely recycled concrete aggregate (RCA) and Type C fly ash, on the overall performance of a special type of pavement surface mixture, porous asphalt mixture. Mixtures of different combinations of RCA (for partial aggregate replacement) and fly ash (for filler replacement) were prepared in the laboratory and tested for a variety of pavement surface performance parameters, including air-void content, permeability, Marshall stability, indirect tensile strength, moisture susceptibility, Cantabro loss, macrotexture, and sound absorption. The analysis of the results showed that incorporating RCA or fly ash in a porous asphalt mixture slightly reduced the air-void content, permeability, and surface macrotexture of the mixture. A 10% replacement of granite aggregates with RCA in the porous asphalt mixtures led to a reduction in mixture stability, indirect tensile strength, resistance to raveling, and sound absorption. The further substitution of mineral filler with fly ash in the mixture, however, helped to offset the negative impact of RCA and brought the mechanical properties of the mixture with 10% RCA to levels comparable to those of the control mixture.

## 1. Introduction

Natural aggregates are a non-renewable resource for the construction of transportation infrastructure and their manufacturing process typically consumes a high amount of energy and creates substantial pollutants and ecological damage. Since aggregates account for over 90 percent of the mass of asphalt pavement, achieving long-term sustainability in asphalt pavement construction and maintenance has become a major challenge in the industry [1]. As one solution, recycled concrete aggregate (RCA) generated from construction and demolition waste may be used to replace the natural aggregates in asphalt pavement construction to improve sustainability and reduce costs. As such, the recycled aggregates not only help to preserve natural resources by reducing the need for quarrying and mining, but also protect the environment by eliminating waste from landfill areas [2]. Another viable technique for establishing a sustainable infrastructure is to incorporate industrial wastes, such as fly ash, into asphalt pavement. Over 65 percent of the fly ash generated by coal-fired power plants is disposed of in landfills around the world [3]. Coal ash is one of the most prevalent types of industrial waste in the United States. According to the American Coal Ash Association, around 130 million tons of coal ash were produced in 2014 [4]. The recycling of fly ash has become a significant concern in recent years, owing to the rising expense of landfill sites and the current interest in sustainable development in the industry.

Recent years have seen a significant increase in the popularity of alternative pavement designs, which are intended to help to mitigate the environmental consequences of civilization’s expansion [5]. For its excellent stormwater runoff reduction function, porous asphalt mixture (PAM) pavements are becoming increasingly widespread in the paving industry [6]. In contrast to impervious pavement, which is considered one of the primary causes of stormwater runoff and flooding in urban areas, PAM allows rainwater to drain at a typical permeability of 0.2 to 0.54 cm/s, making it one of the most environmentally friendly options [7]. In most cases, the porosity formed by the pore network is between 15 and 20 percent or higher, with a porous surface thickness ranging between 6 and 15 cm on average [8]. PAM surfaces have virtually little water films on them when it rains, owing to their high drainage capacity. When PAM is used as a friction course on roadway surface, it helps to reduce the hydroplaning potential of vehicles traveling during rain and thus lowers the risk of wet weather crashes. As an additional benefit, the PAM pavement surface can lower traffic noise in comparison to dense-graded asphalt (DGA) pavement surface [9].

RCA is classified as a two-phase material, in contrast to natural aggregates, which are classified as single phase materials [10]. This means that RCA is made of virgin aggregates and hydrated cement (reclaimed mortar). Since reclaimed mortar is weaker, absorbs more water, and is less resistant to abrasion than most virgin aggregates, RCA is often more absorbent and has lower specific gravity than virgin aggregates [11]. With reduced strength, RCA is suitable for use in pavement base and sub-base layers or in low-traffic road pavement [12]. Currently, its application in hot mix asphalt (HMA) is limited due to its lower quality than natural aggregates, which could have a significant to minor impact on the properties of HMA mixtures [13,14,15,16]. In some previous studies on the use of RCA in asphalt mixtures, it was found that increasing the quantity of RCA in HMA mixtures made the mixtures more sensitive to moisture damage [17,18,19]. The primary reason for the reduced moisture resistance and increased raveling failure in asphalt mixtures containing RCA is a reduction in the mixture tensile strength [19].

Polymer modification can be used to improve the performance of asphalt. This technology, however, is expensive due to the high cost of polymers used in the process [20]. Instead, by incorporating additional byproducts such as fly ash, asphalt mixtures with RCA may have improved strength and moisture resistance. In the literature, previous research on the use of fly ash in asphalt mixtures showed that the addition of fly ash improved the workability of HMA during construction and increased its durability and resistance to water and freeze–thaw damage [21]. It was deemed that the unique spherical shape, good size distribution, and chemical properties of fly ash contributed to the enhanced performance of asphalt mixtures [20]. Although attention has been paid to using fly ash, primarily as a filler, in asphalt mixtures, most work to date has been on the use of fly ash in DGA mixtures, and the application of fly ash in asphalt pavement construction is still limited [22]. There is little work conducted on the application of fly ash in PAM containing RCA.

The objectives of this research are to evaluate the performance of PAM containing RCA and fly ash and to provide recommendations on the design of PAM containing RCA and fly ash. The research approach is based on the laboratory experimental evaluation of PAM performance parameters that are relevant to field pavement functions.

## 2. Materials

In this investigation, two types of aggregates were used. For the control mixtures, crushed granite with ASTM size number 89 and nominal maximum aggregate size of 9.5 mm was used. The second form of aggregate material was RCA. The aggregate gradation for the PAM mix design was selected according to the Federal Highway Administration (FHWA) technical advisory criteria [23], as shown in Figure 1. The selected gradation had 95% of the aggregates passing a 9.5 mm (3/8 in.) sieve, 32% passing a 4.75 mm (No. 4) sieve, 14% passing a 2.36 mm (No. 8) sieve, and 4% passing a 0.075 mm (No. 200) sieve. Granite aggregates of all sizes were involved, but the RCA was chosen from materials that passed the 9.5 mm (3/8 in.) sieve and was retained on the 2.36 mm (No. 8) sieve. The physical and mechanical properties of both types of aggregates were evaluated, and the findings are shown in Table 1. Compared to granite aggregates, the specific gravity of RCA is noticeably lower. The water absorption of RCA is 14.5 times greater than that of granite in the coarse section, which can be attributed to the vast number of pores in the cement mortar adhered to the surface of the RCA. The Los Angeles (LA) abrasion value of RCA was about thrice that of granite. A hybrid mixture was developed to meet the water absorption and abrasion loss limits of utilizing aggregates in HMA for several countries, which are 2% and 25%, respectively [24]. As a result, a maximum of 10% RCA was identified to replace the granite aggregates. Fly ash (Type C), a well-known antistripping agent, was used for filler fraction (<0.075 mm). This type of fly ash was chosen because of its high concentration of CaO (Table 2). This promotes the formation of a strong bitumen-to-aggregate connection, which increases the moisture resistance of asphalt mixtures [25]. The filler impact was evaluated using stone dust from granite stone that passed a 0.075 mm (No. 200) sieve. The specific gravity of fly ash is very similar to that of stone dust, as shown in Table 3. In this study, PG 67-22 asphalt was utilized as a binder for both the control and hybrid specimens, and its technical test results are shown in Table 4. PG 67-22 asphalt is an unmodified asphalt binder that is commonly used in Florida pavement projects [26]. In comparison to modified binders, which are often expensive, this type of asphalt is more cost-effective for low-strength applications.

## 3. Mix Design

The mix design of all PAM samples was performed in accordance with Florida Department of Transportation (FDOT) test method FM 5-588 [31]. This technique uses a visual determination method to establish the optimum binder content (OBC). The process started with placing an uncompacted loose asphalt mixture in a clear glass plate with asphalt contents of 5, 5.5, and 6%. The loose mixture was then heated for about one hour at a temperature of 160 ± 3 °C (320 ± 5 °F). According to FM 5-588 [31], the optimum binder content would cause only the minimal drainage of asphalt binder at the glass plate’s interface with the coated aggregate particles, which was determined to be 6%. PAM mixtures were then divided into four groups to monitor the effects of aggregate types and filler types on the performance of the mixtures, as shown in Table 5. The first two groups are control mixtures N and F, which include 100 percent granite aggregates. The filler type makes the difference between mixtures N and F; fillers account for approximately 4% of the total mass of the mixture proportions. The third and fourth groups (R and O) are hybrid mixtures in which RCA is used to replace 10% of the granite aggregates by mass within a sieve size range of 4.75–2.36 mm and filler type differs as well. Before compaction, the PAM loose mixtures were subjected to short-term conditioning for about 4 h at the temperature of 135 ± 3 °C (275 ± 5 °F). The purpose of this protocol is to simulate the aging of asphalt binders that occurs during plant production and construction as required in AASHTO R 30 [32]. The specimens used in this investigation were prepared using a conventional Marshall compactor. A cylindrical mold with a diameter of 101.6 mm was used to fabricate triplicate specimens for each test. Fifty blows were applied to each side of the specimen during compaction. All the compacted specimens were kept at a room temperature of 25 °C until they were tested.

## 4. Laboratory Tests

### 4.1. Volumetric Properties

The volumetric parameters of the four PAM designs, including theoretical maximum specific gravity (G_mm_), bulk specific gravity (G_mb_), and air-void (A_v_) content, were measured and computed in the laboratory. G_mm_ is defined by ASTM 2041 as a virtual value equivalent to a compressed sample devoid of air spaces [33]. G_mb_ was obtained from the bulk volume measurements using a Parafilm method [34]. Once G_mm_ and G_mb_ were determined, Equation (1) was used to calculate the air-void content (A_v_) in the unit of percentage, which is one of the most essential characteristics of a porous asphalt mixture.
(1)$$A_{v}=\left(1-\frac{G_{\mathrm{mb}}}{G_{\mathrm{mm}}}\right)\times 100$$

### 4.2. Permeability Test

Since PAM is intended to serve as a drainage layer in pavement systems, its permeability is an important parameter to consider. The permeability of each PAM group was determined using a falling head permeability apparatus specifically designed and constructed for this purpose (Figure 2) in accordance with Florida test method FM 5-565 [35]. The coefficient of permeability, k, was calculated using Equation (2).
(2)$$k=\frac{a\;\times L}{A\;\times t}\times \ln\left(\frac{h_{1}}{h_{2}}\right)\times t_{c}$$
where, k = coefficient of permeability (cm/s); A = cross-section area of the sample (cm^2^); L = height of the sample (cm); a = internal cross-section area of the graduated tube (cm^2^); t = time required for water in the graduated tube to fall from h_1_ to h_2_ (s); h_1_ and h_2_ = head at the start and the end of the measurement, respectively (cm); and t_c_ = correction of temperature for water viscosity.

### 4.3. Marshall Stability and Flow Test

When it comes to the Marshall test, PAM material is rarely required to meet any specified requirements because PAM is not considered in pavement design to contribute to the structural load-carrying capacity of the pavement. However, the strength of an asphalt mixture is evaluated in the Marshall test as an experimental indicator for load and deformation resistance [12]. The Marshall stability and flow test was carried out in accordance with AASHTO T 245 specifications [36]. To avoid the significant creep deformation in the PAM specimens during the high temperature conditioning procedure, this test was conducted at 25 °C instead of 60 °C [5,37]. The cylindrical specimen was loaded inside a Marshall breaking head in the diametrical direction to determine the maximum load supported. All specimens were subjected to a constant loading rate of 51 ± 3 mm per minute.

### 4.4. Moisture Susceptibility Test

Because of the considerable amount of air voids inherent in the PAM material, moisture penetration through the mixture is common, leading to water-related distresses. This is especially true in tropical areas with rainy weather, such as south Florida. The modified AASHTO T 283 [38] test procedure described in ASTM D 7064 [39] was used to determine the moisture susceptibility of the four mixtures. In the test, five cycles of freeze and thaw conditioning were performed on the specimens rather than one cycle to better assess the long-term aggregates/asphalt compatibility in the field [39]. The tensile strength ratio (TSR), which is defined as the ratio of the indirect tensile strength of the conditioned specimens and that of the unconditioned specimens, was used to assess the moisture susceptibility of a mixture. Generally, the TSR needs to be greater than 80% for the mixture to be acceptable to have sufficient moisture resistance [39].

### 4.5. Cantabro Test

Raveling is the most visible and costly problem that can occur on PAM pavements [40]. The resistance of a PAM mixture to raveling is commonly evaluated in the laboratory by the Cantabro test [41]. This test is performed at 25 °C using a Los Angeles abrasion machine by placing PAM cylindrical specimens in the drum chamber without steel balls and rotating the drum for 300 revolutions at a speed between 30 and 33 revolutions per minute. The mass of a specimen before and after the drum revolutions is measured and used to calculate the percentage abrasion loss (AL), as shown in Equation (3).
(3)$$\mathrm{AL}=\left(\frac{W_{1}-W_{2}}{W_{2}}\right)\times 100$$
where AL = abrasion loss value (%); W_1_ = sample’s original mass; and W_2_ = sample’s final mass.

### 4.6. Macrotexture

The macrotexture of the pavement has a significant impact on the skid resistance and tire/pavement noise [42]. Because of this, understanding the texture of the pavement is critical for pavement management purposes. To overcome the limitations of the prior volumetric approach used to compute the Mean Profile Depth (MPD), a laser texture scanner from KEYENCE called wide-area 3D measurement system (VR-5000) was utilized to obtain high-accuracy 3D images of the surface texture of the specimens. The VR-5000 has a wide mapping area of up to 200 × 100 × 50 mm (7.87′′ × 3.94′′ × 1.97′′) and a precision of the height difference down to 1 µm. Three samples from each surface type were tested to compute the MPD from the surface profile. A baseline of 100 mm was established for each sample, and the software automatically computed the profile’s average level. Each specimen was divided into 12 segments, 2 of which served as the baseline, as shown in Figure 3. Six baselines across the specimen’s center were chosen randomly for MPD computations. The maximum profile peak height (Rp) parameter was utilized to produce a peak level for the first and second segments. This parameter reflects the maximum peak height of a profile within the baseline. A noise filtration was also applied to remove the peaks greater than 2.5 mm. MPD was calculated by Equation (4).
(4)$$\mathrm{MPD}=\left(\frac{\mathrm{Peak}\;\mathrm{level}\;\left(1\mathrm{st}\right)-\mathrm{Peak}\;\mathrm{level}\;\left(2\mathrm{nd}\right)}{2}\right)$$

### 4.7. Measurement of Sound Absorption

The sound absorption coefficient (α) of cylindrical PAM specimens was evaluated using a low-cost impedance tube built specifically for this experiment, as illustrated in Figure 4. Both European standard ISO 10534-2 [43] and US standard ASTM E-1050 [44] were followed to build the impedance tube. In this method, a 101.6 mm (4 inches) in diameter transparent plastic tube has a speaker at one end and a specimen at the other. At one end, a cylindrical cap filled with foam is placed on the impedance tube opposite to the PAM specimen position to reduce speaker noise. A cylindrical cap with a 101.6 mm diameter piston is attached to the other end of the specimen to create a rigid end for signal reflection purposes. A software program named Audacity is used to generate a sound pulse (white noise), which is amplified by an amplifier and sent via the speaker into the impedance tube [45]. The sound then travels until it reaches the specimen, where some sound energy is absorbed and the rest is reflected by a steel plate attached to the rear of the specimen. Two microphones connected to the sound interface record the incident and reflected sound wave amplitudes. A MATLAB-based software program (A|Lab) is used to calculate the material’s average sound absorption coefficient [46].

## 5. Test Result and Analysis

### 5.1. Optimum Binder Content

Figure 5 shows the bottoms of 228.6 mm (9 inches) in diameter pie plates with the uncompacted asphalt mixtures of three trial binder contents (5.5%, 6%, and 6.5%) and the same aggregate/RCA/filler combination. Following the FDOT test method FM 5-588 [31], the OBC was determined visually to be the one at which the sample showed appropriate bonding between the mixture and the bottom of the pie plate without indicating excessive asphalt binder draindown. For all four PAM designs included in this study, their OBCs were all determined to be 6% by mass of aggregate.

### 5.2. Air-Void Content and Permeability Properties

The air-void contents of the four mixtures are summarized in Figure 6 in terms of the average values and the range of one standard deviation. It can be seen from the figure that the average air-void contents of the four mixtures ranged from 22.7 to 23.5 percent, which is deemed within the desired limit recommended in the ASTM D 7064 specification for PAM, which is 18 percent or above [39]. Both RCA and fly ash tend to reduce the air-void content of PAM when comparing the mixture F (containing fly ash as the filler type) and the mixture R (containing 10% RCA) to the mixture N (containing 100% granite aggregates and stone dust as the filler), while the combination of fly ash and RCA does not lead to a further reduction in the air-void content, as shown in the mixture O (containing both RCA and fly ash). In addition, the average change in the mixture air-void content (from 22.76 to 23.5%) is minor from an engineering application perspective.

Figure 7 illustrates the average coefficients of permeability of the four mixtures along with the one standard deviation range. From the figure, it can be seen that the permeability of the compacted specimens is greater than 100 m/day for all four mixtures, which is the desirable minimum level for PAM [39]. Additionally, these data revealed the same trend in the air-void content results, with the permeability of the mixture N being the highest and somewhat decreasing when RCA or fly ash was included. However, it did not appear to be a significant difference in permeability between the mixtures containing RAC, fly ash, or a combination of both.

### 5.3. Marshall Stability and Flow

The Marshall stability and flow experiment was performed to evaluate the strength and stiffness of the asphalt mixtures. A parameter named Marshall Quotient (MQ) is derived from the results of the stability and flow test as a ratio between load (stability) and deformation (flow). This parameter serves as an indicator of the stiffness of the mixture: the higher the ratio is, the stiffer and more cohesive the mixture becomes [47]. As shown in Figure 8, when fly ash was used as a filler, there was a minor to moderate improvement in the stability of the mixtures F and O. However, when comparing the two hybrid mixtures (R and O) in terms of MQ, a significant improvement in stiffness was observed, with the stiffness of the hybrid mixture O increasing by over 15% when compared to the hybrid mixture R, which contains stone dust as a filler. Due to the much rougher and more porous surfaces of RCA compared to granite particles, fly ash proved to have superior performance in filling the voids between the aggregate particles and the surface pores of RCA, resulting in a greater stability as compared to stone dust as the filler [48].

### 5.4. Moisture Susceptibility

Figure 9 shows the results of the moisture susceptibility test. As can be seen, replacing 10% granite aggregates with RCA reduced the average indirect tensile strength of PAM from 0.90 MPa to 0.77 MPa. With the use of fly ash as the filler in the mixture, however, the negative impact of RCA on the indirect tensile strength was completely offset. For the mixtures after moisture conditioning, the combination of RCA and fly ash helped to improve the indirect tensile strength, as can be seen when comparing the results from mixtures O and N.

In terms of TSR, although none of the groups met the minimum value requirement of 80%, the hybrid groups had the best results in resisting the moisture damage, likely owing to the rougher RCA surface texture and the chemistry of fly ash, which influence the bonding between the recycled aggregates and asphalt mastic. Similar findings were reported in the literature [49,50]. Asphalt mixtures containing granite aggregates and stone dust (N) were shown to be more vulnerable to moisture damage than the hybrid aggregates group. The replacement of the stone dust filler with fly ash for the hybrid mixtures resulted in improvement in TSR by almost 10%. It should be noted that granite aggregates are well known for its poor bonding performance with asphalt in the existence of moisture, and the addition of antistrip additives, such as liquid antistripping agents and hydrated lime, may further bring the mixture TSR beyond 80%.

### 5.5. Resistance to Raveling 

The Cantabro test was used in this study to determine the PAM resistance to raveling. Figure 10 presents the average and standard deviation of the Cantabro loss values obtained from the four groups of mixtures. As can be seen in the figure, the average Cantabro loss for the four mixtures was around 40%. The granite mixture (F), which contains fly ash, was found to be within the same range of Cantabro loss as that of the hybrid mixture (O), which contains the same type of filler. However, the hybrid mixture (R), which contains stone dust as a filler, showed the highest loss value of about 45%.

### 5.6. Macrotexture

Figure 11 summarizes the mean profile depth (MPD) values measured using the wide-area 3D measurement system (VR-5000). Despite all the mixtures used in this study having the same aggregate gradation, differences in surface macrotexture occurred due to the use of different aggregate and filler types. MPD is slightly higher for mixtures containing granite aggregates than for mixtures using hybrid aggregates. The use of RCA in mixture R resulted in a decrease in the MPD value of around 32%. This could be due to the fact that mixtures with natural aggregates are more difficult to compact, resulting in a less densified mixture, which leads to higher MPD values [51]. The use of fly ash as the filler in PAM also reduced the macrotexture from 1.47 mm for mixture N to 1.21 mm for mixture F, as shown in the figure. The average MPD values of the hybrid mixtures (R and O), on the other hand, were determined to be 1.09 and 1.06 mm, respectively. These values are below the typical MPD range (1.5 to 3 mm) of open-graded HMA [52], but still fall in the “very good” category for general pavement surfaces as defined in the uniform European performance indicators for pavements [53].

### 5.7. Sound Absorption

Figure 12 illustrates the average sound absorption coefficient measured at different frequencies for the four groups of mixtures. Granite mixtures N and F were shown to absorb almost 80% of the sound energy at the low-frequency range between 200 and 600 Hz. However, at the frequency of 1400 Hz, all mixtures demonstrated a significant reduction in sound absorption of up to 20% to 30%. Furthermore, the presence of RCA in mixtures R and O resulted in a generally lower sound absorption coefficient when compared to mixtures N and F. Nonetheless, when comparing the two hybrid mixtures, it appears that the fly ash inclusion in mixture O led to better low-frequency sound absorption characteristics than mixture R.

## 6. Discussion

From the laboratory test results presented in the previous section, it can be seen that the use of RCA and fly ash in porous asphalt mixtures had various impacts on different mixture properties. In general, both materials slightly increased the mixture density and thus led to reductions in air-void content, coefficient of permeability, and surface macrotexture. The impact of RCA may be explained by the fact that RCA particles had rough surfaces and sharp edges with weaker reclaimed mortar and thus were more prone to breaking and crushing during mixture compaction. The degradation of RCA increased the number of fine particles, which may fill up more voids in the mixture. The impact of fly ash may be explained by its unique features of spherical shape, good size distribution, and chemical properties, which helped to improve the rheological properties and workability of the hot mix asphalt [20,48].

This study selected 10% as the maximum replacement rate of granite aggregates by RCA to meet the requirements of water absorption and abrasion loss of aggregates in asphalt mixtures. Nevertheless, there was still a noticeable negative impact of RCA on the mechanical and durability properties of the porous asphalt mixtures, which was expected. Specifically, at 25 °C, the average Marshall stability dropped from 27.0 kN to 24.8 kN; the average indirect tensile strength dropped from 0.90 MPa to 0.77 MPa; and the average Cantabro loss increased from 38% to 44%. The reduction in the mixture stability and tensile strength was mainly due to the low strength of the RCA particles. The increase in the Cantabro loss also suggests that RCA particles have less resistance to mechanical degradation and their surface may have weaker bonding with asphalt than granite aggregates. The further substitution of stone dust with fly ash, however, brought the mechanical properties of the mixture with 10% RCA to the levels comparable to those of the control mixture. Specifically, the average Marshall stability, indirect tensile strength, and Cantabro loss were changed to 27.8 kN, 0.90 MPa, and 37%, respectively. This shows the beneficial effects of coupling two industrial waste/byproduct materials in the porous asphalt mixtures.

## 7. Conclusions

This study explored the potential of using recycled concrete aggregate (RCA) and fly ash in porous asphalt mixtures for the main objective of reducing the negative environmental impact of human activities. One porous asphalt mixture with granite aggregates and PG 67-22 binder was selected as the control mixture, and its partial granite aggregates and stone dust filler were replaced with RCA and fly ash, respectively. The laboratory evaluation of the various performance indices of the mixtures led to the following conclusions:The RCA’s water absorption and abrasion loss were greater than those of the granite aggregates. As a result, a maximum of 10% replacement of granite aggregates with RCA was determined to meet the aggregate absorption and abrasion limits in the asphalt mixture design.Incorporating either RCA or fly ash in a porous asphalt mixture may promote the densification of the mixture during compaction and lead to a slightly lower air-void content and permeability that still meet the requirements of the porous asphalt mixture for pavement surface performance. The macrotexture of the compacted porous asphalt mixture surface was also reduced by RCA or fly ash by up to 28%.The 10% replacement of the granite aggregates with RCA in the porous asphalt mixtures also reduced the overall sound absorption performance of the mixture. Nonetheless, using fly ash as the filler in the mixture containing RCA resulted in an improved sound absorption performance comparable to that of the control mixture.The 10% replacement of granite aggregates with RCA in porous asphalt mixtures led to a reduction in mixture stability, tensile strength, and resistance to raveling. The further replacement of stone dust with fly ash, however, helped to offset the negative impact of RCA and brought mixture stability, tensile strength, and raveling resistance back to the levels of the control mixture.The 10% replacement of granite aggregates with RCA in the porous asphalt mixtures improved the mixture resistance to moisture damage and using fly ash as the filler further improved the moisture resistance.

## 8. Future Work

This study used the recycled concrete aggregate materials obtained from one source. Considering that the physical properties of RCA may vary greatly depending on the material sources and manufacturing techniques, further work may be performed with RCAs from other sources to determine appropriate RCA contents to be recommended for porous asphalt mixtures. Additional mixture performance variables, such as resistance to rutting and fatigue cracking, may also be evaluated for porous asphalt mixtures containing RCA and fly ash to be placed in pavements with high heavy traffic volumes.

## Figures and Tables

**Figure 1 materials-15-06363-f001:**
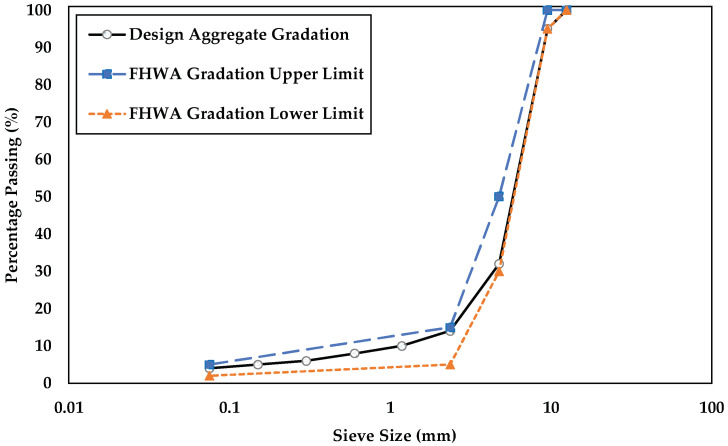
Design aggregate gradation to meet FHWA specification requirements for PAM.

**Figure 2 materials-15-06363-f002:**
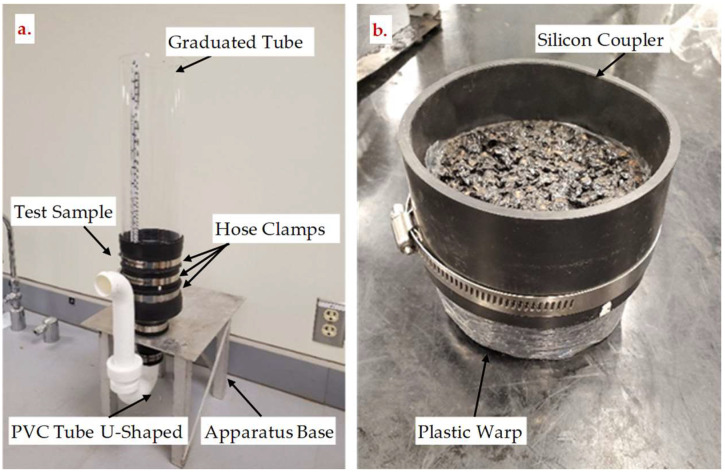
(**a**) Falling head permeability apparatus (**left**); (**b**) test sample preparation (**right**).

**Figure 3 materials-15-06363-f003:**
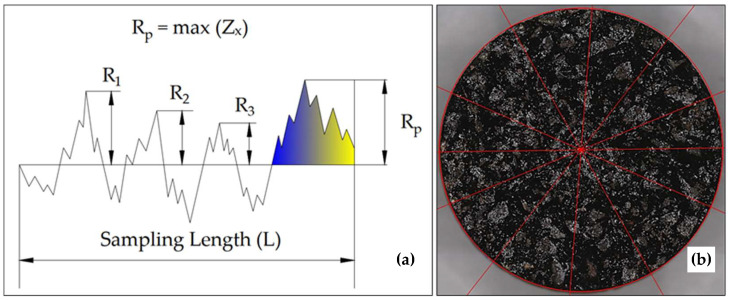
(**a**) Maximum profile peak height (Rp) (**left**); (**b**) PAM specimen with divided segments (**right**).

**Figure 4 materials-15-06363-f004:**
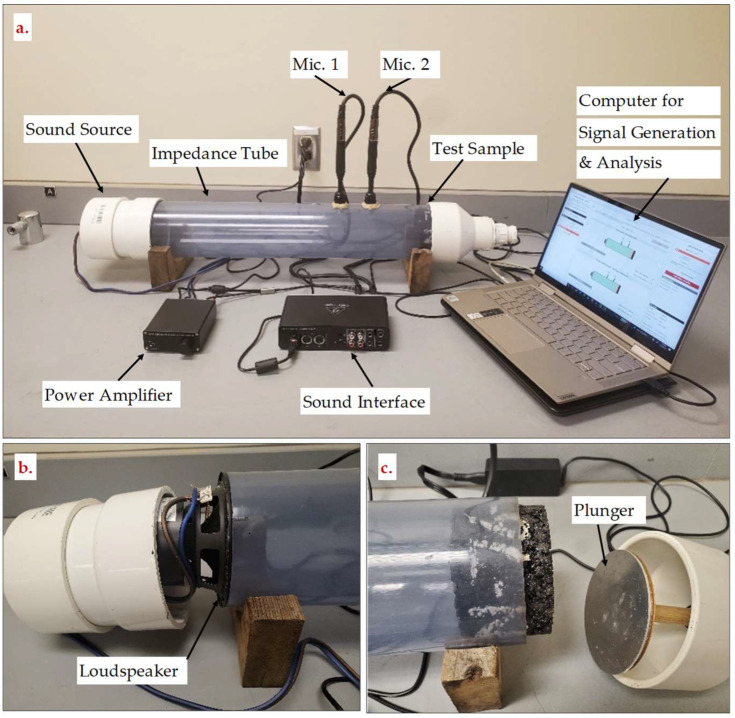
(**a**) Impedance tube used for the sound absorption testing; (**b**) loudspeaker end; (**c**) test sample and hard-backing end.

**Figure 5 materials-15-06363-f005:**
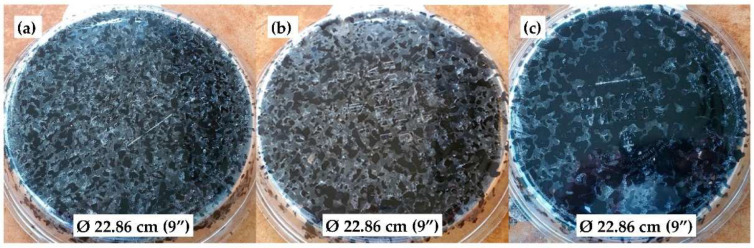
Samples used to determine the OBC using the pie plate method [31]: (**a**) 5.5% binder content; (**b**) 6.0% binder content; and (**c**) 6.5% binder content.

**Figure 6 materials-15-06363-f006:**
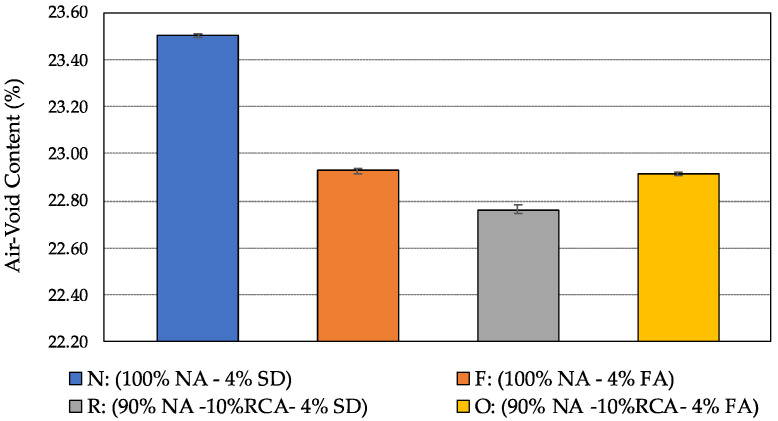
Average air-void content of the four porous asphalt mixtures.

**Figure 7 materials-15-06363-f007:**
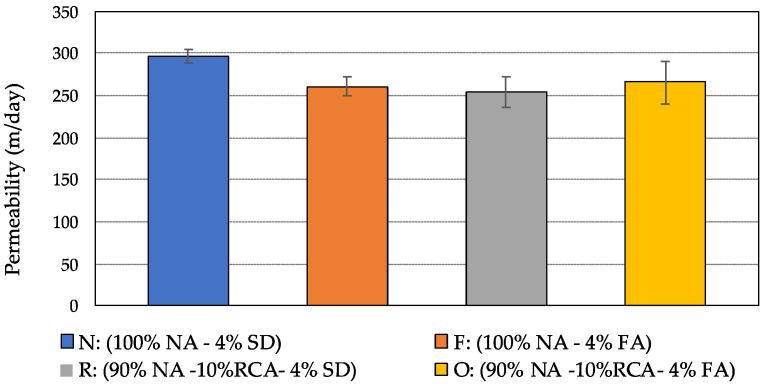
Average permeability coefficient of the four porous asphalt mixtures.

**Figure 8 materials-15-06363-f008:**
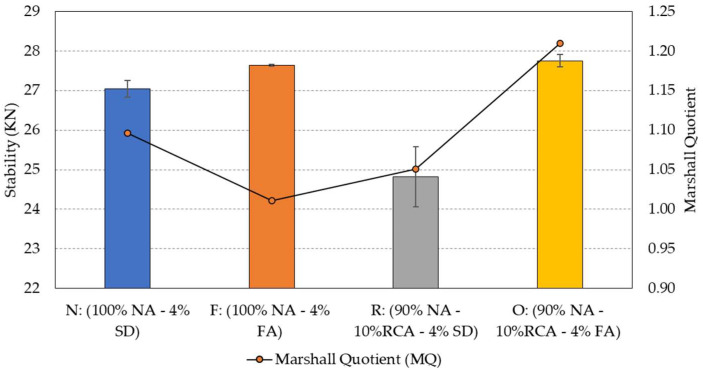
Average Marshall stability and Marshall Quotient at 25 °C of the four porous asphalt mixtures.

**Figure 9 materials-15-06363-f009:**
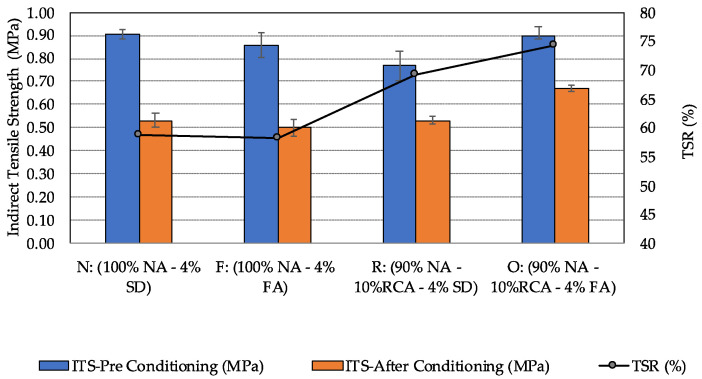
Average indirect tensile strength and TSR at 25 °C of the four porous asphalt mixtures.

**Figure 10 materials-15-06363-f010:**
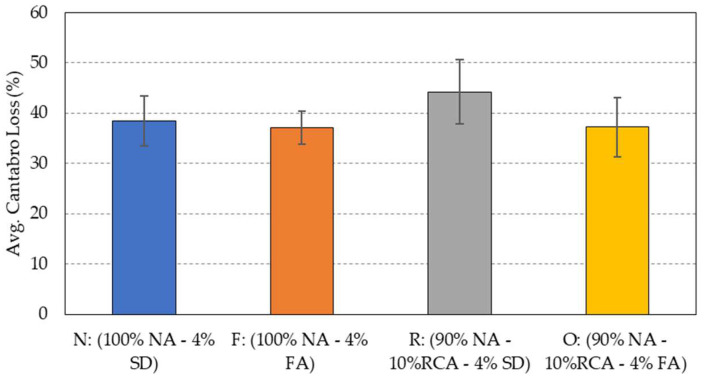
Average Cantabro loss of the four porous asphalt mixtures at 25 °C.

**Figure 11 materials-15-06363-f011:**
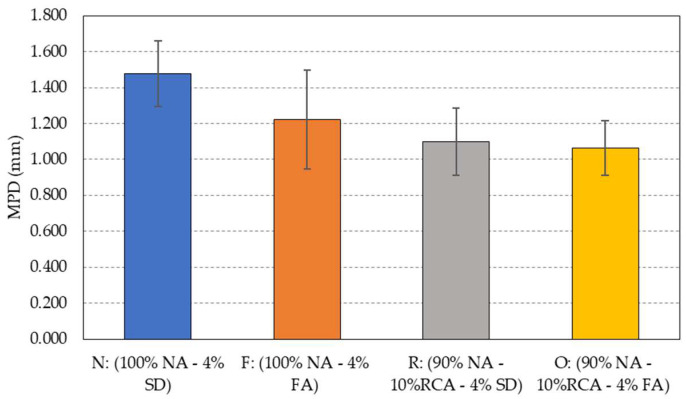
Average MPD of the four porous asphalt mixtures.

**Figure 12 materials-15-06363-f012:**
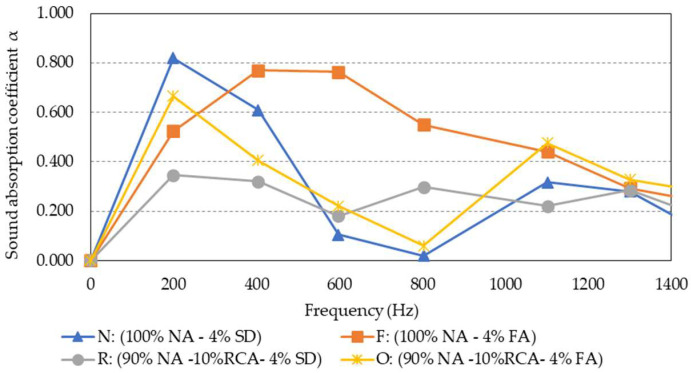
Average sound absorption coefficient of the four porous asphalt mixtures.

**Table 1 materials-15-06363-t001:** Properties of granite and RCA.

Property	Test Method	Aggregate Type		
		Granite (NA)	RCA	Hybrid
Los Angeles Abrasion, %	ASTM C 131 [27]	19	45	24.5
Absorption, %	ASTM C 127 [28]	0.7	10.2	1.49
Specific Gravity (Dry)	ASTM C 127 [28]	2.741	2.081	2.561
Specific Gravity (SSD)	ASTM C 127 [28]	2.761	2.281	2.599
Specific Gravity (Apparent)	ASTM C 127 [28]	2.796	2.603	2.663
Solid Waste or Hazardous Materials, %	AASHTO MP 16 [29]	-	-	-
Metal, Wood, Plaster, and Gypsum, %	AASHTO MP 16 [29]	-	0.5	0.05
Brick Content, %	AASHTO MP 16 [29]	-	2	0.2
Bituminous Concrete Content, %	AASHTO MP 16 [29]	-	3	0.3

**Table 2 materials-15-06363-t002:** Chemical composition of fly ash type C (provided by material supplier).

Raw Materials	CaO_2_	SiO_2_	Al_2_O_3_	Fe_x_O_y_	MgO	SO_3_	Na_2_O
Fly Ash	23.3	38.14	17.88	6.16	4.18	2.17	2.93

**Table 3 materials-15-06363-t003:** Specific gravity of fillers.

Property	Test Method	Filler Type	
		Stone Dust (SD)	Fly Ash (FA)
Specific Gravity	ASTM C 128 [30]	2.67	2.64

**Table 4 materials-15-06363-t004:** Properties of asphalt binder PG 67-22 (provided by material supplier).

Properties	Test Results
Dynamic Shear (G*/sin δ, 10 rad/s), kPa	1.17
Specific Gravity @ 15.6 °C	1.031
Density @15.6 °C, kg/l	1.029
API Gravity @ 15.6 °C, API	5.71
Rotational Viscosity, Pa·s @ 135C, 20 rpm, Spindle 21 @ 165C, 20 rpm, Spindle 21	0.472 0.128
Flash Point Standard, °C	366
Penetration @ 25 °C, 0.1 mm	54
Ring and Ball Softening Point, °C	50.56

**Table 5 materials-15-06363-t005:** PAM designs included in the study.

Sieve Size Range (mm)	Mass Proportion (%)	Experimental Group				
		Binder Content (%)	Granite Mix 1 (N)	Granite Mix 2 (F)	Hybrid Mix 3 (R)	Hybrid Mix 4 (O)
9.5–4.75	5	6	Granite	Granite	Granite	Granite
4.75–2.36	63	6	Granite	Granite	90% Granite + 10% RCA	90% Granite + 10% RCA
2.36–0.075	28	6	Granite	Granite	Granite	Granite
<0.075	4	6	Granite	Fly Ash	Granite	Fly Ash

## Data Availability

The data presented in this study are available upon request from the corresponding author.

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
