# Peer review of "Performance of Porous Asphalt Mixtures Containing Recycled Concrete Aggregate and Fly Ash"

_materials, 2022, doi:10.3390/ma15186363_

Round 1
Reviewer 1 Report
Title: Laboratory investigation of the performance of porous asphalt mixtures…
Manuscript ID: materials-1884044
Authors: Elmagarhe et al.
Dear Authors,
Thank you for the opportunity to read your article. I found the topic is interesting and fundamental. Generally speaking, there are some results presented in order to capture some trends but the methods and results need more clear explanation and detail discussion with fair point of view. I suggest that this article will be revised extensively before its re-submission for another review process if applicable. As a conclusion, I recommend its major revision at this state.
I hope my comments are helpful.
Good luck,
A reviewer
Major concerns:
“Abstract”
-Line 19: “RCA”->Please provide the full spelling of RCA.
Please consider your abstract is a stand-alone content.
“Keywords”
->Please consider providing keywords that are not used in the article title.
“1. Introduction”
-Introduction is too long and contains too general information not directly relevant to your specific study. Please consider limiting your introduction and focus on the topics directly relevant to your study.
-Lines 106-107: “To date, no studies have been done on the application of fly ash in PAM containing RCA.”->Please consider revising this statement as it is highly suspicious.
-Based on your literature review, in the introduction, please consider clearly stating research gap(s) you tried to address in this study. In other words, please mention why you studied “to determine the influence of a small percentage of RCA as well as fly ash Type C in a PAM on the overall performance of the mixture…to determine if a PAM containing RCA and/or fly ash has volumetric properties and performance comparable to a mixture that does not contain RCA or fly ash…to make appropriate recommendations on mixture design in general.”
“2. Research Objectives”
->Please consider combining this section with the introduction as it has only 1 paragraph.
“3. Materials and Mix Design”
“3.1 Materials”
-Figure 1: (a) Please consider explaining the results more. (b) Please consider revising the statement on lines 123 to 124 “RCA…passed through a 9.5 mm sieve and were retained on a 4.75 mm sieve” that is against your results showing the presence of minus 4.75 mm.
“3.2 Mix Design”
-Lines160-162: “The appropriate binder level would be suggested when the asphalt mixture contains drainage but not too much on the container…”->(a) Please consider stating a quantity with literature instead of this vague definition if possible. (b) Please also justify why you used a fixed binder content for different particle size ranges if I understand from Table 5. Particle size usually affects the porosity (similar to drainage in your definition) of particle packing.
“4. Laboratory Tests”
-Figure 5 (and elsewhere applicable): (a) Please consider adding sub-figure numbers (e.g., a, b, c) and explaining them in the figure title. (b) Please consider pointing out which part of images represent which part of the equipment (e.g. electrode?) or specimen.
“5. Test results analysis and discussion”
-It would be great if you could provide a general summary of all the test results at the end of this section, and suggest the best/better alternative of the conventional mixture. From the information/results given in this current section, it is not easy to understand your suggestion(s) but the section is just a gather of all the results.
-Figure 6 (and elsewhere): In the figure title, please introduce (a) what N, F, R, and O stand for and (b) all the key experimental conditions to produce your samples and to test them.
-Lines 294-295: “…due to the fact that fly ash produces a stiffer mixture…”->(a) Please consider showing the fact you have other than the results shown in Fig.6, or revising your statement. (b) Please consider explaining why fly ash produces a stiffer mixture.
Minor concerns:
-Please consider polishing English more. You may use some of my comments above and below for this purpose.
-Lines 28-29: “Natural aggregates are a non-renewable raw resource that is…”-> Natural aggregates are non-renewable raw resources that are…
Author Response
Reviewer 1:
Major concerns:
“Abstract”
-Line 19: “RCA”->Please provide the full spelling of RCA. Please consider your abstract is a stand-alone content.
Response: The complete definition of RCA has been added in line 13 as follows: "recycled concrete aggregate (RCA)"
“Keywords”
->Please consider providing keywords that are not used in the article title.
Response: The key words have been updated to the following: “porous asphalt mixture; recycled concrete aggregate; fly ash; mix design; macrotexture; Cantabro test; sound absorption”
“1. Introduction”
-Introduction is too long and contains too general information not directly relevant to your specific study. Please consider limiting your introduction and focus on the topics directly relevant to your study.
Response: Thank you for the invaluable comment. We have revised the introduction section and removed information that we considered unrelated to the topic of the study.
-Lines 106-107: “To date, no studies have been done on the application of fly ash in PAM containing RCA.”->Please consider revising this statement as it is highly suspicious.
Response: We agree with the reviewer that this is a strong statement. We have conducted a further literature review and found little information on previous studies that looked at the application of fly ash in PAM containing RCA. We have revised the statement to “There is little work done on the application of fly ash in PAM containing RCA.”
-Based on your literature review, in the introduction, please consider clearly stating research gap(s) you tried to address in this study. In other words, please mention why you studied “to determine the influence of a small percentage of RCA as well as fly ash Type C in a PAM on the overall performance of the mixture…to determine if a PAM containing RCA and/or fly ash has volumetric properties and performance comparable to a mixture that does not contain RCA or fly ash…to make appropriate recommendations on mixture design in general.”
Response: We have significantly revised the introduction section to provide discussion and justification that lead to our research objectives. We have also revised our research objectives as follows:
“The objectives of this research are to evaluate the performance of PAM containing RCA and fly ash and to provide recommendations on the design of PAM containing RCA and fly ash.”
“2. Research Objectives”
->Please consider combining this section with the introduction as it has only 1 paragraph.
Response: We have merged this section into the Introduction section.
“3. Materials and Mix Design”
“3.1 Materials”
-Figure 1: (a) Please consider explaining the results more.
Response: The results shown in Figure 1 are explained from line 93 to line 97 as follows: “The aggregate gradation for the PAM mix design was selected according to the Federal Highway Administration (FHWA) technical advisory criteria, as shown in Figure 1. The selected gradation has 95% of the aggregates passing a 9.5 mm (3/8 in.) sieve, 32% passing a 4.75 mm (No. 4) sieve, 14% passing a 2.36 mm (No. 8) sieve, and 4% passing a 0.075 mm (No. 200) sieve.”
(b) Please consider revising the statement on lines 123 to 124 “RCA…passed through a 9.5 mm sieve and were retained on a 4.75 mm sieve” that is against your results showing the presence of minus 4.75 mm.
Response: The statement has been revised as shown from line 97 to 98 as follows: “Granite aggregates of all sizes were involved, but the RCA was chosen from materials that passed the 9.5 mm (3/8 in.) sieve and was retained on the 4.75 mm (No. 4) sieve.”
“3.2 Mix Design”
-Lines160-162: “The appropriate binder level would be suggested when the asphalt mixture contains drainage but not too much on the container…”->(a) Please consider stating a quantity with literature instead of this vague definition if possible.
Response: The statement has been revised as shown from line 253 to 256 as follows: “Following the FDOT test method FM 5-588, the OBC was determined visually to be the one at which the sample showed appropriate bonding between the mixture and the bottom of the pie plate without indicating excessive asphalt binder draindown.” Since the FM 5-588 test method determines the optimum binder content based on visual assessment, there is no quantity used as a criterion for the optimum binder content. Currently, there is some research effort trying to use image analysis technique along with machine learning to come up with a more objective approach to determine the optimum binder content, but the research work has not resulted in implementable results yet.
(b) Please also justify why you used a fixed binder content for different particle size ranges if I understand from Table 5. Particle size usually affects the porosity (similar to drainage in your definition) of particle packing.
Response: the same binder content used for the four mixtures included in the study was determined based on the FM 5-588 visual assessment approach as explained in the response to the previous comment. Since the replacement rate of granite aggregates by RCA is low (10%) and the content of mineral filler in the aggregate gradation replaced by fly ash is also low (4%), they did not seem to cause a significant change in the optimum binder content as determined based on FM 5-588.
“4. Laboratory Tests”
-Figure 5 (and elsewhere applicable): (a) Please consider adding sub-figure numbers (e.g., a, b, c) and explaining them in the figure title.
Response: The sub-figure numbers have been added and explained in the figure titles of Figure 5 and other relevant figures.
(b) Please consider pointing out which part of images represent which part of the equipment (e.g. electrode?) or specimen.
Response: The arrow pointing system has been added to explain various parts of the equipment in the images.
“5. Test results analysis and discussion”
-It would be great if you could provide a general summary of all the test results at the end of this section, and suggest the best/better alternative of the conventional mixture. From the information/results given in this current section, it is not easy to understand your suggestion(s) but the section is just a gather of all the results.
Response: We have added a discussion section (Section 6) to summarize and discuss our findings from the tests included in the study.
-Figure 6 (and elsewhere): In the figure title, please introduce (a) what N, F, R, and O stand for and
Response: Explanations of the letters (N, F, R, and O) have been added to all relevant figures.
(b) all the key experimental conditions to produce your samples and to test them.
Response: The titles of the figures have been revised to reflect the key experimental conditions in figures (8,9 &10).
-Lines 294-295: “…due to the fact that fly ash produces a stiffer mixture…”->(a) Please consider showing the fact you have other than the results shown in Fig.6 or revising your statement.
Response: We have deleted this statement.
(b) Please consider explaining why fly ash produces a stiffer mixture.
Response: One possible reason is the high amorphous glass components (SiO2 and Al2O3) in fly ash make it chemically active and bond easily with asphaltic acid and anhydride, as discussed in the literature. However, since our tests did not directly address the stiffening effect of fly ash on asphalt mixture, we have deleted the relevant statement as explained in the response to the previous comment.
Minor concerns:
-Please consider polishing English more. You may use some of my comments above and below for this purpose. -Lines 28-29: “Natural aggregates are a non-renewable raw resource that is…”-> Natural aggregates are non-renewable raw resources that are…
Response: A detailed review was carried out in order to refine the manuscript's English language writing.

Reviewer 2 Report
The manuscript entitled "Laboratory Investigation of the Performance of Porous Asphalt Mixtures Containing Solid Aggregate and Filler Wastes" presents an interesting experimental study conducted on the obtaining and characterization of asphalt with RCA and fly ash addition. However, the scientific organization of the paper is questionable and many other issues must be addressed. The paper needs major revisions before it is processed further, some comments follow:
Title: The title is too long and unclear. Please consider replacing the title with a clear formula that reflects the content of the manuscript. "Laboratory investigation of the performance" can be removed.
Abstract: Please remove the unnecessary acronyms from the Abstract. This section must be suitable for separate presentations (independent of the manuscript text body). Therefore, please keep only the acronyms that are used in the abstract and introduce the acronyms for the manuscript the first time when those terms appear in the manuscript text body.
The abstract is written qualitatively. The majority of the qualitative statements should be modified for quantified result comparisons. Please add some quantitative comparisons related to the following sentences:
"lead to a slightly lower", "led to a reduction in mixture", "helped offset the negative". Also, please provide some short formulations related to the results obtained for each test.
Introduction Section
The introduction section can be improved, since there are no references to study from 2022, and the number of references/studies from the last five years is low.
The authors stated that "There is currently a limit to the use of recycled... ...mixtures [15–17]." But the cited studies are 10 years old and do not reflect the current state of this matter. Please refer to the data from these studies (DOI: 10.3390/ma15113929, ISBN: 9780323898386) make clear evaluation and statements in the introduction section of your manuscript and cite these publications.
Please remove the research objective section and provide the novelty statement in the last paragraph of the introduction: Please highlight the novelty of the study in accordance with previous publications.
Materials and Methods section
Table 1 - two types of iron oxides have been detected in this type of material, therefore, please replace Fe2O3 with FexOy or provide scientific proof to support your results.
Figure 2 – Please introduce a scale bar on the figures and introduce figure labels to highlight the areas on interest for the readers.
4. Laboratory Tests – this section should be integrated into the materials and methods section since it contains the description of the involved methods/equipment.
Figure 3. Please introduce figure labels and indicate the components of the designed apparatus.
Figure 5 – same comments as for Figure 3.
When laboratory-designed equipment are involved in the research, the authors should provide all the information/data necessary for experiment repeatability.
The results presented in this section should be moved to the Results section.
Results and discussions
The results were mixed with the methods section. Please clearly separate/divide the methods from the results section.
Discussion section. The discussion section is missing. In the discussions section, clear correspondence and comparison between the results of this study and those from the literature should be provided. Please improve.
Future directions and limitations: Please provide some future directions and limitations of the study. This section is very important for this study.
Author Response
Reviewer 2:
The manuscript entitled "Laboratory Investigation of the Performance of Porous Asphalt Mixtures Containing Solid Aggregate and Filler Wastes" presents an interesting experimental study conducted on the obtaining and characterization of asphalt with RCA and fly ash addition. However, the scientific organization of the paper is questionable and many other issues must be addressed. The paper needs major revisions before it is processed further, some comments follow:
Title: The title is too long and unclear. Please consider replacing the title with a clear formula that reflects the content of the manuscript. "Laboratory investigation of the performance" can be removed.
Response: We have revised the title to a shorter version as following: “Performance of Porous Asphalt Mixtures Containing Recycled Concrete Aggregate and Fly Ash”
Abstract: Please remove the unnecessary acronyms from the Abstract. This section must be suitable for separate presentations (independent of the manuscript text body). Therefore, please keep only the acronyms that are used in the abstract and introduce the acronyms for the manuscript the first time when those terms appear in the manuscript text body.
The abstract is written qualitatively. The majority of the qualitative statements should be modified for quantified result comparisons. Please add some quantitative comparisons related to the following sentences:
"lead to a slightly lower", "led to a reduction in mixture", "helped offset the negative". Also, please provide some short formulations related to the results obtained for each test.
Response: Thank you for your invaluable comments! We have removed unnecessary acronyms in the abstract and only kept one (RCA) in it. The definition of RCA has also been added in the abstract. We tend not to include quantitative statements for all the tests because they are specific to the combinations of materials used in our study and they will significantly increase the length of the abstract. We instead presented the key findings in a more general sense that readers may quickly comprehend. A summary of quantitative statements of the test results was added in the Discussion section.
Introduction Section
The introduction section can be improved, since there are no references to study from 2022, and the number of references/studies from the last five years is low.
The authors stated that "There is currently a limit to the use of recycled... ...mixtures [15–17]." But the cited studies are 10 years old and do not reflect the current state of this matter. Please refer to the data from these studies (DOI: 10.3390/ma15113929, ISBN: 9780323898386) make clear evaluation and statements in the introduction section of your manuscript and cite these publications.
Please remove the research objective section and provide the novelty statement in the last paragraph of the introduction: Please highlight the novelty of the study in accordance with previous publications.
Response: We have significantly reorganized and rewritten the Introduction section. The studies that were cited in the statement and were older than ten years have been removed, and we have performed additional literature review focusing on the publications in the recent decade, including those recommended by the reviewer. We have also combined the research objective section with the introduction section, with the justification of our research objective explained in Lines 81 through 84.
Materials and Methods section
Table 1 - two types of iron oxides have been detected in this type of material, therefore, please replace Fe2O3 with FexOy or provide scientific proof to support your results.
Response: Thank you for pointing this out. The composition information of fly ash was provided by the material supplier. We have revised Fe2O3 to FexOy as recommended by the reviewer.
Figure 2 – Please introduce a scale bar on the figures and introduce figure labels to highlight the areas on interest for the readers.
Response: We have amended the images to display the complete diameter of the clear pie plate, which is 22.86 cm (9"). The figures have also been labeled to reflect the reviewer's recommendation.
- Laboratory Tests – this section should be integrated into the materials and methods section since it contains the description of the involved methods/equipment.
Response: We tend to keep three separate sections for Materials, Mix Design, and Laboratory Tests in the manuscript as combining them into one section will lead to multiple levels of sub-headings.
Figure 3. Please introduce figure labels and indicate the components of the designed apparatus.
Figure 5 – same comments as for Figure 3.
Response: We have revised the figures to label the components of the designed apparatus.
When laboratory-designed equipment is involved in the research, the authors should provide all the information/data necessary for experiment repeatability.
Response: We have added all the necessary information, including citations to the standard specifications based on which our test devices were developed.
The results presented in this section should be moved to the Results section.
Response: We have re-organized the sections to move the discussion of optimum binder content to the “Test Results & Analysis” section.
Results and discussions
The results were mixed with the methods section. Please clearly separate/divide the methods from the results section.
Response: We have re-organized the sections to move the discussion of optimum binder content to the “Test Results & Analysis” section.
Discussion section. The discussion section is missing. In the discussions section, clear correspondence and comparison between the results of this study and those from the literature should be provided. Please improve.
Response: We have added a discussion section (Section 6) to summarize and discuss our findings from the tests included in the study.
Future directions and limitations: Please provide some future directions and limitations of the study. This section is very important for this study.
Response: We have added a section (Section 8) to address the limitation of our work scope and to present future work directions.

Round 2
Reviewer 1 Report
Dear Authors,
As all the comments were addressed, I would suggest the journal accept this article for its publication.
Best regards,
A reviewer
Reviewer 2 Report
The authors adressed all of my comments and improved the paper accordingly. The article can be processed further.